

# Four sesquiterpene glycosides from loquat (*Eriobotrya japonica*) leaf ameliorates palmitic acid-induced insulin resistance and lipid accumulation in HepG2 Cells via AMPK signaling pathway

Jiawei Li[1,*], Xiaoqin Ding[1,*], Tunyu Jian[1], Han Lü[1], Lei Zhao[1], Jing Li[2], Yan Liu[1], Bingru Ren[1] and Jian Chen[1,2]

[1] Institute of Botany, Jiangsu Province and Chinese Academy of Sciences, Nanjing, China
[2] Department of Food Science and Technology, College of Light Industry and Food Engineering, Nanjing Forestry University, Nanjing, China
[*] These authors contributed equally to this work.

Corresponding author
Jian Chen, chenjian80@aliyun.com

## ABSTRACT

Insulin resistance (IR), caused by impaired insulin signal and decreased insulin sensitivity, is generally responsible for the pathophysiology of type 2 diabetes mellitus (T2DM). Sesquiterpene glycosides (SGs), the exclusive natural products from loquat leaf, have been regarded as potential lead compounds owing to their high efficacy in hypoglycemia and hypolipidemia. Here, we evaluated the beneficial effects of four single SGs isolated from loquat leaf, including SG1, SG2, SG3 and one novel compound SG4 against palmitic acid-induced insulin resistance in HepG2 cells. SG1, SG3 and SG4 could significantly enhance glucose uptake of insulin-resistant HepG2 cells at non-cytotoxic concentration. Meanwhile, Oil Red O staining showed the decrease of both total cholesterol and triglyceride content, suggesting the amelioration of lipid accumulation by SGs in insulin-resistant HepG2 cells. Further investigations found that the expression levels of phosphorylated AMPK, ACC, IRS-1, and Akt were significantly up-regulated after SGs treatment, on the contrary, the expression levels of SREBP-1 and FAS were significantly down-regulated. Notably, AMPK inhibitor Compound C (CC) blocked the regulative effects, while AMPK activator AICAR mimicked the effects of SGs in PA-treated insulin-resistant HepG2 cells. In conclusion, SGs (SG4>SG1≈SG3>SG2) improved lipid accumulation in insulin-resistant HepG2 cells through the AMPK signaling pathway.

## INTRODUCTION

Diabetes mellitus (DM) has become a heterogeneous global epidemic characterized by chronic hyperglycemia associated with impaired lipids metabolism (*Motahari-Tabari et al., 2014*). Currently, there are 463 million diabetic patients, and the number is estimated

to reach 700 million by 2045 (*Saeedi et al., 2019*). Most of DM patients are affected by type 2 diabetes mellitus (T2DM) which is mainly due to the role of defects with insulin resistance (*Fonseca, 2009*; *Jaiswal et al., 2017*). Insulin resistance is characterized by the weak sensitivity of insulin in peripheral tissues such as skeletal muscle, adipose tissue and liver, resulting hyperglycemia, hyperinsulinemia and dyslipidemia. As the vital organ for metabolic homeostasis and main target organ of insulin action, liver controls the regulation of glucose and lipid metabolism. Hepatic insulin resistance leads to severely dysregulated glucose homeostasis and lipid over-accumulation, which would aggravate hepatic insulin insensitivity(*Samuel & Shulman, 2012*). Therefore, improving hepatic insulin resistance could be an effective strategy in the prevention of T2DM and diabetes-related diseases.

AMP-activated protein kinase (AMPK) is an important kinase that has a critical influence on regulating energy metabolism (*Herzig & Shaw, 2018*). Studies have found that AMPK as a key master switch for regulating fatty acid oxidation, triglyceride, lipogenesis, cholesterol synthesis and gluconeogenesis in the liver (*Liu, Deng & Fan, 2019*). Activation of AMPK was proved to decrease cholesterol, plasma glucose and triglycerides production in the liver (*Hasenour, Berglund & Wasserman, 2013*). Phosphorylated AMPK level was down-regulated in the diabetic liver, and activation of AMPK mediated the reverse of metabolic disorders and alleviation of liver function in diabetes mice (*Chen et al., 2020*; *Liu, Deng & Fan, 2019*). Thus, AMPK may be an attractive target for insulin resistance in T2DM. With low toxicity and high efficiency, natural products have been used as alternative anti-T2DM agents. Several natural products have been demonstrated to enhance glucose uptake and attenuate insulin resistance via activating the AMPK pathway (*Mazibuko-Mbeje et al., 2019*; *Wang et al., 2017*; *Yan et al., 2018*).

The fruit tree, loquat (*Eriobotrya japonica*), is a small evergreen arbor and its leaf is an important traditional herbal medicine capable of counteracting inflammation, diabetes, and cancer (*Liu et al., 2016*). It has been reported that the chemical constituents of loquat leaf included phenolics (*Fu et al., 2020*), flavonoids (*Lü et al. 2009*), terpenoids (*Zhang et al., 2019*), essential oils (*Hong et al., 2010*), sesquiterpene glycosides (SGs) (*Ao et al., 2015*) and polysaccharides (*Lu et al., 2019*). So far, the pharmacological activities of SGs in loquat were mainly manifested as hypoglycemic and anti-nonalcoholic fatty liver disease (NAFLD). Hypoglycemic activity of SGs which was studied in a C57BL/ks-db/db/Ola hereditary diabetic mouse model showed that one of the SGs produced a marked inhibition of glycosuria (*De Tommasi et al., 1991*). In addition, SGs could significantly reduce the blood glucose level of alloxan-diabetic mouse model, while it had no significant effect on normal mice (*Chen et al., 2008*). Total SGs have been proven to alleviate oxidative stress and NAFLD in high-fat diet induced mouse model and oleic acid induced cell model of NAFLD (*Jian et al., 2017*; *Jian et al., 2018*).

Our previous investigations on the phytochemistry of loquat leaf led to the discovery of six SGs (*Ao et al., 2015*; *Chen et al., 2008*; *Zhao et al., 2015*). To our knowledge, the influence of SGs on glucose uptake and lipid accumulation in insulin-resistant HepG2 cells have not been investigated. In this study, three known SGs,

nerolidol-3-$O$-$\alpha$-L-rhamnopyranosyl-(1 →4)-$\alpha$-L-rhamnopyranosyl-(1 →2)-[$\alpha$-L-rhamnopyranosyl-(1 →6)]-$\beta$-d-glucopyranoside (SG1) (*Chen et al., 2008*), nerolidol-3-$O$-$\alpha$-L-rhamnopyranosyl-(1 →4)-$\alpha$-L-rhamnopyranosyl-(1 →2)-$\beta$-d-glucopyranoside (SG2) (*De Tommasi et al., 1991*) and nerolidol-3-$O$-$\alpha$-L-rhamnopyranosyl-(1 →2)-[$\alpha$-L-rhamnopyranosyl-(1 →6)]- $\beta$-d-glucopyranoside (SG3) (*Ao et al., 2015*), together with a new compound named as nerolidol-3-O-$\alpha$-L-arabinopyranosyl-(1 →4)-$\alpha$-L-rhamnopyranosyl-(1 →2)-[$\alpha$-L-rhamnopyranosyl-(1 →6)]-$\beta$-d-glucopyranoside (SG4), were isolated and structural elucidated from loquat leaf. Given the importance of hepatic functions in the pathogenesis of insulin resistance and T2DM, we used a palmitic acid (PA)-induced insulin resistant (IR) model in HepG2 cells to evaluate the efficacy of these four SGs and tried to elucidate the underlying possible molecular mechanism in vitro.

## MATERIAL AND METHODS

### Plant material and reagents

The loquat leaf, collected from Xishan island, in Suzhou city of Jiangsu Province of China, was identified by Prof. Bingru Ren. A voucher specimen (No. 328636) was deposited in the Herbarium of the Institute of Botany, Jiangsu Province and the Chinese Academy of Sciences.

SGs (purity ≥ 98%, HPLC) were prepared in our laboratory. PA was purchased from Macklin (Shanghai, China). Fetal bovine serum (FBS) and Dulbecco modified Eagle medium (DMEM) were obtained from Gibco (Carlsbad, CA, USA). AMPK activator AICAR and inhibitor Compound C (CC) were obtained from Beyotime Institute of Biotechnology (Haimen, China) and Selleck (Houston, TX, USA), respectively. The primary antibodies: anti-phospho-IRS-1 (Tyr895) (#3070S), anti-phospho-Akt (#9271), anti-IRS-1 (#2382S), anti-Akt (#9272), anti-$\beta$-actin (#4970) were obtained from Cell Signaling Technology (Danvers, MA, USA), anti-phospho-AMPK (T183/172) and anti-AMPK (D168) were purchased from Bioworld (Bloomington, MN, USA), anti-FAS (10624-2-AP) and anti-ACC (21923-1-AP) were purchased from Proteintech Group (Chicago, USA), and anti-p-ACC (sc-271965) and anti-SREBP-1 (sc-365513) were purchased from Santa Cruz Biotechnology (Santa Cruz, CA, USA). HRP-linked anti-mouse (#7076) and anti-rabbit (#7074) secondary antibodies were obtained from Cell Signaling Technology (Danvers, MA, USA). CCK-8 assay and BCA Protein Quantification Kit were purchased from Biosharp (Hefei, China). The Oil Red O stain kit (For Cultured Cells) was obtained from Solarbio (Beijing, China). Total cholesterol (TC), triglyceride (TG) and glycogen assay kits were obtained from Nanjing Jiancheng Bioengineering Institute (Nanjing, China). 2-[N-(7-nitrobenz-2-oxa-1,3-diazol-4-yl) amino]-2-deoxy-D-glucose (2-NBDG) was purchased from Life Technologies (Carlsbad, CA, USA). 10% SDS-PAGE was obtained from EpiZyme Biotechnology (Shanghai, China). Polyvinylidene difluoride membranes were purchased from Millipore (Millipore, MA, USA). High-sig chemiluminescence (ECL) reagent was purchased from Tanon (Shanghai, China). EtOH and the other reagents were obtained from Sinopharm Chemical Reagent Co., Ltd.

## Extraction and isolation

The method for extraction and isolation of SGs were as follows: the dried leaf of loquat (10 kg) was pulverized into powder and percolated with 80% EtOH for two months at room temperature. The combined extract was evaporated under reduced pressure to remove alcohol, afterward, the extracts were centrifuged (3000 g for 15 min), and the supernatants were column chromatographed over macroporous resin (XAD16) eluting with 0, 40%, 60%, 70%, and 95% EtOH, respectively. The 60% and 70% EtOH eluted fractions were combined and further column chromatographed over polyamide eluting with different ultrapure $H_2O$: EtOH mixtures (10:0, 7:3, 5:5, 3:7, 0:10), respectively. The $H_2O$ elution fraction was collected and column chromatographed by RP-C18 with $H_2O$: MeOH mixtures (9:1, 7:3, 5:5, 4:6, 3:7, 8:2, 0:10) as solvent to obtain a total of SGs compounds (1.02 g). The final purification of SG1 (203.0 mg), SG2 (13.8 mg), SG3 (158.6 mg) and SG4 (12.4 mg) were achieved by preparative HPLC (LC-6AD, Shimadzu) on a RP-C18 column (5 $\mu$m, 9.4 × 250 mm, Agilent) with 65∼70% MeOH. The NMR spectrum was obtained from a Bruker Avance III 400 MHz spectrometer in DMSO-$d_6$. HR-ESI-MS spectrum was recorded on a 6530 UPLC-Q-TOF mass spectrometer (Agilent, USA). The acid hydrolysis of SG1-4 and their sugar analysis were carried out as described previously (*Ao et al., 2015*).

## Cell culture

HepG2 cells were obtained from the Cell Bank of Shanghai Institute of Cell Biology, Chinese Academy of Sciences. The cells were cultured in DMEM supplemented with 10% FBS and 1% antibiotics (penicillin and streptomycin) in a humidified atmosphere with 5% $CO_2$ at 37 °C.

## Cell Counting Kit-8 (CCK-8) assay

The cell viabilities were assessed using CCK-8 assay. In brief, HepG2 cells were plated into 96-well plates with $2 \times 10^5$ per well and incubated overnight. Afterwards, the medium was changed with DMEM containing different concentrations of SGs for another 24 h. Subsequently, CCK-8 working solution was added to each well and cultivated for another 1 h. The absorbance was recorded on Microplate reader at 450 nm (Molecular Device, Sunnyvale, USA).

## Induction of IR HepG2 cells model

The IR model was induced in HepG2 cells with 0.25 mM PA (*Heo et al., 2018*). PA-bovine serum albumin (BSA) conjugate was prepared by dissolving in 50% ethanol and mixing in aqueous BSA solution, and finally diluted in culture media. The cells cultured with BSA only served as the control. HepG2 cells were pre-treated with or without 250 $\mu$M PA for 24 h, then incubated in the culture media with or without 250 $\mu$M PA or SGs (5 and 10 $\mu$M, dissolved in ultrapure water) for another 24 h. Control media was prepared to contain equivalent amount of ethanol and BSA.

## Measurement of Glucose Uptake and glycogen content

The uptake of 2-NBDG in HepG2 cells were measured as follows: HepG2 cells were seeded in 96-well plates at a concentration of $2 \times 10^5$ per well, then to induce IR and intervened

with SGs as described above. Subsequently, the cells were incubated in glucose-free DMEM with 100 nM insulin for 30 min, and then they were incubated with 100 μM 2-NBDG for another 30 min or harvested. The fluorescent intensity was measured on a multimode microplate reader (Berthold TriStar LB941, Germany) at 485 nm excitation and 535 nm emission wavelengths. Harvested cells were used to detect glycogen content according to the manufacturer's instruction.

## Measurement of lipid uptake in HepG2 cells

The cells were homogenized in lysis buffer. The levels of intracellular TC and TG were detected with commercial assay kits following the manufacturer's instruction. The data were normalized against protein concentration.

The total lipid content in HepG2 cells was measured by Oil red O staining. Briefly, the cells were fixed in 4% formaldehyde for 15 min and then cleaned with PBS, stained with Oil Red O working solution for 30 min. After immediately washed with 60% isopropanol, incubated with hematoxylin for 5 min and washed by PBS, the cells were immediately imaged using microscopy (Olympus, Tokyo, Japan), and quantified by ImageJ version 1.52a software ((NIH, USA).

## Western blot analysis

After treatment as above, the total proteins of HepG2 cells were obtained through cell lysis buffer containing 0.1 mM of phenylmethanesulfonyl fluoride. Then the cell lysates were centrifuged (12,000 g for 15 min) at 4 °C. Supernatants were prepared and the protein concentration was assayed with BCA Protein Quantification Kit according to the manufactory's instructions. Protein from the samples was separated by SDS-PAGE, transferred to PVDF membranes and then blocked with 5% fat-free milk and incubated at 4 °C overnight. The membrane was incubated with primary antibody at 4 °C for 24 h, then washed with Tris Buffered saline Tween (TBST) followed by incubating the blot with secondary antibody. Membranes were visualized using the enhanced chemiluminescence (Tanon, China). The band intensities were analyzed using ImageJ version 1.52a software (NIH, USA).

## Statistical analysis

The experimental data was performed as the means $\pm$ SD. Multiple comparisons were done by One-way analysis of variance (ANOVA) followed by Dunnett's post hoc test using GraphPad Prism 8. Data were considered statistically significant at a $P$-value of less than 0.05.

## RESULTS

### Identification of SG4

SG4 was obtained as a white powder. It was identified to have a molecular formula of $C_{38}H_{64}O_{18}$ on the basis of the high-resolution electrospray ionization-time of flight mass spectrometry ($m/z$ 807.5577 [M-H]$^-$). SG4 was assumed to be a sesquiterpene glycoside since its physicochemical properties and spectral characteristics (Table 1). In the $^1$H NMR

spectrum, $\delta_H$ 5.21 (H-1a, 1H), 5.17 (H-1b, 1H), 5.76 (H-2, 1H), 5.08 (H-6, 1H) and 5.07 (H-10, 1H) could be assigned to a vinyl proton. The proton signals at $\delta_H$ 1.56 (H-12, 3H) and 1.63 (H-15, 3H) exhibited a gem-dimethyl group on olefinic carbons. The singlet at $\delta_H$ 1.54 (H-14, 3H) indicated a methyl next to a double bond, and the singlet at $\delta_H$ 1.27 (H-13, 3H) was deduced to be another methyl group on a quaternary carbon. The four multiple peaks, each integrating two protons at $\delta_H$ 1.46 (H-4), 1.92 (H-5), 1.94 (H-8) and 2.01 (H-9), could be inferred from the relevant methylenes. In $^{13}$C NMR spectrum, the signals at $\delta_C$ 116.1 (C-1), 143.4 (C-2), 124.7 (C-6), 134.7 (C-7), 124.6 (C-10) and 131.1 (C-11) were assigned to three pairs of olefinic carbon signals. The signal at $\delta_C$ 80.0 (C-3) showed one oxygenated carbon as well. By comparison of values data in previous reference (*De Tommasi et al. 1990*), it was suggested that the aglycone of SG4 was nerolidol.

The downfield shift in the $^{13}$C NMR spectrum of SG4 demonstrated that it was glycosylated at the C-3 position. In the sugar portion, four protons at $\delta_H$ 4.25 (H-1′, 1H), 5.13 (H-1″, 1H), 4.32 (H-1‴, 1H), and 4.56 (H-1⁗, 1H) in $^1$H NMR spectrum were examined to be correlated with their sugar anomeric carbons at $\delta_C$ 96.9 (C-1′), 100.2 (C-1″), 106.2 (C-1‴) and 101.3 (C-1⁗) in HMQC spectrum, respectively. According to pre-column derivatization GC analysis of authentic monosaccharides, the monosaccharides of SG4 were determined to be D-glucose, L-rhamnose and L-arabinose (ratio1:2:1). The signal at $J_{H-1′,H-1′}$ (7.6 Hz) indicated that the $\beta$-anomeric configuration of the glucose, and the resonances of C-3 ($\delta_C$ 71.0, 71.1) and C-5 ($\delta_C$ 66.8, 68.8) demonstrated that the $\alpha$-anomeric configurations of the rhamnose units, and the signal at $J_{H-1‴, H-2‴}$ (7.5 Hz) suggested that the $\alpha$-anomeric configuration of arabinose (Kasai et al. 1979). HMBC spectrum correlations showed that the following sequence of the sugar linkages connected to the aglycone: H-1″($\delta_H$ 5.13) of rhamnose I with C-2′($\delta_C$ 78.5) of glucose, H-1‴($\delta_H$ 4.32) of arabinose with C-4″($\delta_C$ 83.7) of rhamnose I H-1′($\delta_H$ 4.25) of glucose with C-3 ($\delta_C$ 80.0) of aglycone, and H-1⁗($\delta_H$ 4.56) of rhamnose II with C-6′($\delta_C$ 67.5) of glucose (Table 1). Therefore, SG4 was analyzed to be one new compound, and it was characterized as nerolidol-3-O- $\alpha$-L-arabinopyranosyl-(1 →4)-$\alpha$-L-rhamnopyranosyl-(1 →2)-[$\alpha$-L-rhamnopyranosyl-(1 →6)]-$\beta$-d-glucopyranoside (Fig. 1).

## The effects of SGs on HepG2 cells viability

The cytotoxic effect of a series concentration of SGs (50-250 μM) in HepG2 cells was evaluated using CCK-8 assay after 24 h incubation. As shown in Fig. 2, the viability of cells treated with 250 μM of SG1, 200 μM and 250 μM of SG2 were all significantly reduced, accordingly, both SG3 and SG4 at concentration of 250 μM significantly reduced cells viability as well. These results showed that SGs displayed distinct growth inhibition when drug concentrations were more than 200 or 250 μM, indicating that 5 or 10 μM concentrations of SGs for the present study were safe on HepG2 cells.

## The effects of SGs on glucose uptake and glycogen content in IR HepG2 cells

Studies pointed out that glucose uptake assay in IR cells determined whether compounds could ameliorate IR in HepG2 cells induced by PA (*Tong et al., 2018*). As shown in  Figs. 3A

Li et al. (2020), *PeerJ*, DOI 10.7717/peerj.10413

**Table 1** $^1$H and $^{13}$C NMR Data of SG4 (DMSO-d$_6$, δ, ppm, J/Hz).

| C atom | $\delta_C$ | $\delta_H$ | HMBC | | C atom | $\delta_C$ | $\delta_H$ | HMBC |
|---|---|---|---|---|---|---|---|---|
| 1 | 116.1 | 5.21 (1H) 5.17 (1H) | C-3, C-2 | Glc | 1′ | 96.9 | 4.25 (1H, d, $J = 7.6$ Hz) | C-3 |
| 2 | 143.4 | 5.76 (1H, dd, $J = 17.0$, 10.0 Hz) | C-13 | | 2′ | 78.5 | 3.28 | C-3′ |
| 3 | 80.0 | | | | 3′ | 76.3 | 3.17 | C-2′, C-1″, C-1′ |
| 4 | 41.6 | 1.46 (m, 2H) | C-3, C-13 | | 4′ | 69.9 | 3.26 | C-3′ |
| 5 | 22.5 | 1.92 (m, 2H) | C-6, C-4, C-7 | | 5′ | 75.6 | 3.12 | |
| 6 | 124.7 | 5.08 (submerged, 1H) | | | 6′ | 67.5 | 3.77 (1H, m), 3.33 (1H, m) | C-1⁗ |
| 7 | 134.7 | | | Rha I | 1″ | 100.2 | 5.13 (brs, 1H) | C-5″, C-2″, C-3″ |
| 8 | 39.7 | 1.94 (m, 2H) | C-7, C-9, C-14, C-6 | | 2″ | 71.2 | 3.40 | C-3″ |
| 9 | 26.7 | 2.01 (m, 2H) | C-11, C-10, C-8, C-7 | | 3″ | 71.0 | 3.59 | C-2″ |
| 10 | 124.6 | 5.07 (submerged, 1H) | C-8, C-9, C-12, C-15 | | 4″ | 83.7 | 3.37 | C-6″, C-1‴, C-5″, C-3″ |
| 11 | 131.1 | | | | 5″ | 66.8 | 4.00 | C-4″ |
| 12 | 18.0 | 1.56 (s, 3H) | C-11, C-10, C-15 | | 6″ | 18.0 | 1.13 (d, $J = 6.1$ Hz) | C-4″, C-5″ |
| 13 | 22.3 | 1.27 (s, 3H) | C-2, C-3, C-4 | Ara | 1‴ | 106.2 | 4.32 (d, $J = 7.5$ Hz, 1H) | C-4″ |
| 14 | 16.2 | 1.54 (s, 3H) | C-7, C-6, C-8 | | 2‴ | 77.1 | 3.10 | C-3″ |
| 15 | 25.9 | 1.63 (s, 3H) | C-11, C-10, C-12 | | 3‴ | 75.0 | 2.99 | C-1‴ |
| | | | | | 4‴ | 70.4 | 3.67 | |
| | | | | | 5‴ | 66.5 | 3.68, 3.03 | C-1‴ |
| | | | | Rha II | 1⁗ | 101.3 | 4.56 (s, 1H) | C-2⁗, C-5⁗ |
| | | | | | 2⁗ | 71.2 | 3.40 | C-3⁗ |
| | | | | | 3⁗ | 71.1 | 2.96 | |
| | | | | | 4⁗ | 72.5 | 3.18 | C-6⁗, C-3⁗, C-2⁗ |
| | | | | | 5⁗ | 68.8 | 3.41 | C-4⁗ |
| | | | | | 6⁗ | 18.4 | 1.13 (d, $J = 6.1$ Hz, 3H) | C-5⁗, C-4⁗ |

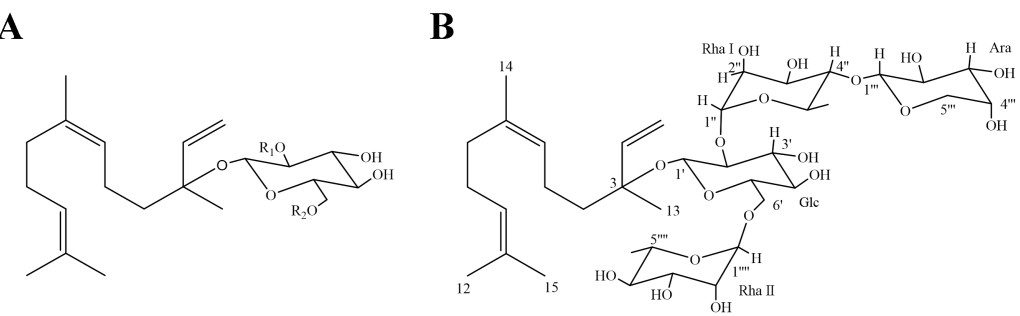

**Figure 1** **Chemical structures of SGs.** (A) SG1: $R_1$ = Rha (1 →4) Rha, $R_2$ = Rha; SG2: $R_1$ = Rha (1 →4) Rha, $R_2$ = H; SG3: $R_1$ = Rha, $R_2$ = Rha; (B) Chemical structure of SG4* (* as a novel compound).

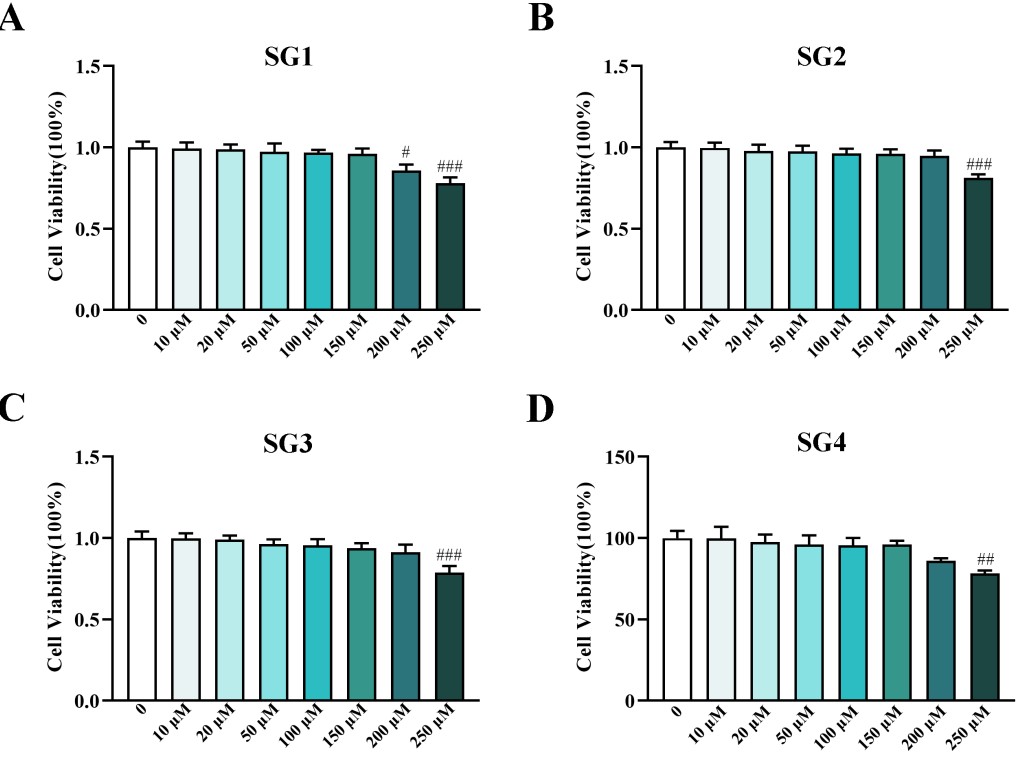

**Figure 2** **Influence of SGs on cell viability in HepG2 cells.** (A, B, C and D) Cell viability of HepG2 cells in different concentrations of SG1, SG2, SG3 and SG4 from 0 to 250 μM ($n = 8$). # $P < 0.05$, ## $P < 0.01$, ### $P < 0.001$ compared with the control HepG2 cells.

and 3B, both the glucose uptake and glycogen content of IR group were significantly lower than that of control group after 24 h treatments. On the contrary, a remarkable increase of glucose uptake and glycogen content were detected in SGs-treated group (except for SG2) in comparison with the IR group.

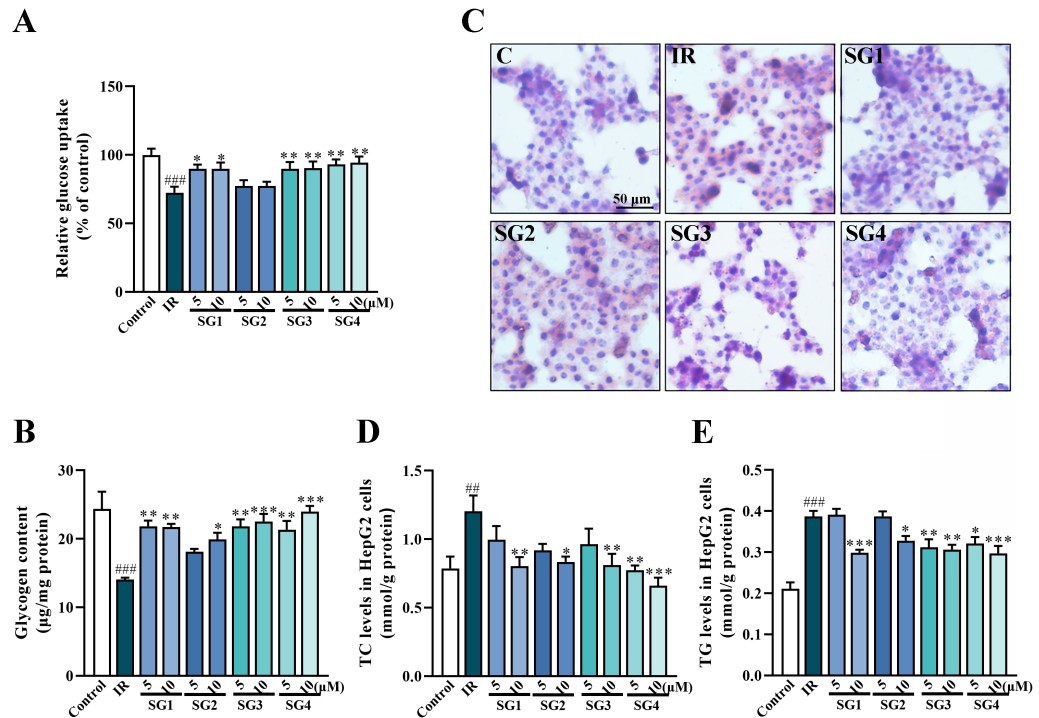

**Figure 3  SGs on insulin-stimulated glucose uptake, glycogen content and lipid deposition in PA-treated HepG2 cells.** (A) Glucose uptake in HepG2 cells measured by 2-NBDG method ($n = 8$); (B) Glycogen content in HepG2 cells; (C) Oil Red O stain of HepG2 cells; (D and E) SGs treatment decreased TC and TG content in PA-treated HepG2 cells ($n = 8$). ## $P < 0.01$ and ### $P < 0.001$ compared with control group; * $P < 0.05$, ** $P < 0.01$, *** $P < 0.001$ compared with the IR group.

## The effects of SGs on lipid accumulation in IR HepG2 cells

The Oil Red O stain method was used to evaluate lipid accumulation in insulin-resistant HepG2 cells. As shown in Fig. 3C, compared with control group, insulin-resistant HepG2 cells had much more lipid droplets, while the SGs-treatments (except for SG2) caused a significant decrease in lipid droplets. These results were further confirmed by the contents of TC and TG in the cells. As shown in Figs. 3D and 3E, in comparison with control group, the level of TC and TG increased significantly after PA inducing, on the contrary, the SGs-treated groups caused a significant decrease.

## The effects of SGs on AMPK signaling pathways in IR HepG2 cells

To investigate underlying possible molecular mechanism of SGs ameliorating lipid accumulation and IR in PA-induced HepG2 cells, we investigated the effects of SGs on the protein expression of the AMPK pathway. As shown in Fig. 4, in comparison with control group, PA-treatment significantly decreased the phosphorylation levels of AMPK, ACC, IRS1 and Akt, and up-regulated SREPB-1 and FAS protein level. However, in comparison with IR group, phosphorylation levels of AMPK, ACC, IRS-1 and Akt were significantly up-regulated in SGs-treated groups, especially in high concentration (10 μM) of SG4, while SG2 had no significant effects on the expressions of p-Akt and p-IRS-1.

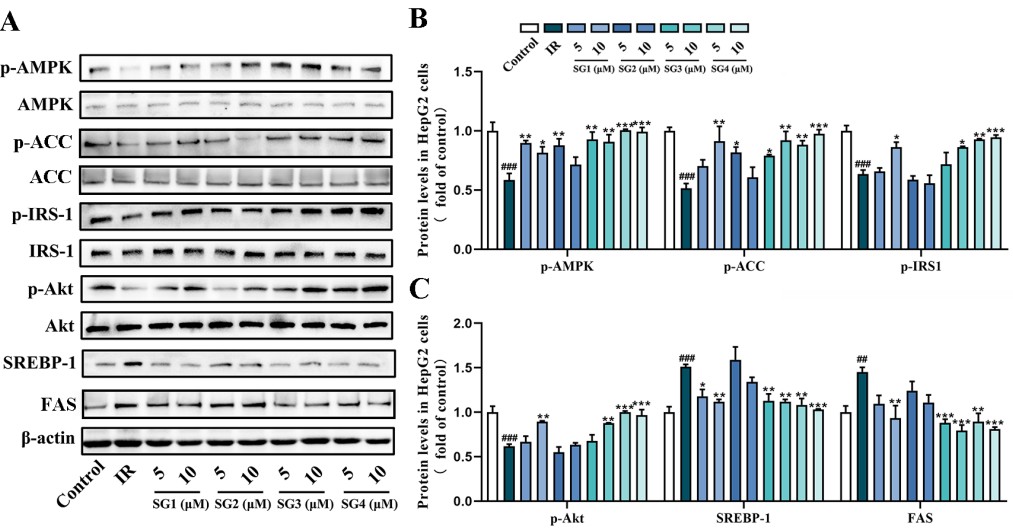

**Figure 4 Western blot analysis of protein expression levels of AMPK signaling pathway.** (A) Expression strips of all protein; (B) Phosphorylation level of AMPK normalized by AMPK, phosphorylation level of ACC normalized by ACC and phosphorylation level of IRS-1 normalized by IRS-1 ($n = 3$); (C) Phosphorylation level of Akt normalized by Akt ($n = 3$), SREBP-1 and FAS protein levels normalized by $\beta$-actin ($n = 3$). ## $P < 0.01$ and ### $P < 0.001$ compared with the control group; * $P < 0.05$, ** $P < 0.01$, *** $P < 0.001$ compared with the IR group.

Moreover, in comparison with IR group, SGs-treated groups (except for SG2) showed a significant reduction of SREBP-1 and FAS level.

To confirm whether SGs regulates AMPK to activate IRS1/Akt insulin signal pathway and decrease SREBP-1/FAS pathway, AMPK activator AICAR and AMPK inhibitor CC were used in PA-treated HepG2 cells. The results showed that AICAR (0.5 mM) up-regulated phosphorylation levels of ACC and IRS1, and reduced SREBP-1 protein level in PA-treated HepG2 cells. However, CC (10 μM) partly abolished, the up-regulative effect of SGs on phosphorylation levels of ACC and IRS1, and the down-regulative effect of SGs on SREBP-1 protein level (Fig. 5). These results suggested that AMPK pathway was involved in the SGs-mediated IRS1 and SREBP-1 pathways in PA-treated HepG2 cells.

## DISCUSSION

As an important traditional Chinese medicine, loquat leaf has been widely used with beneficial effects in numerous diseases including chronic bronchitis, asthma and diabetes (*Fu et al., 2019*; *Kim, Paudel & Kim, 2020*; *Li et al., 2004*; *Zhang et al., 2019*). Our previous study has confirmed the alleviation of hepatic lipid accumulation and hyperlipidemia of total SGs from loquat leaves in high-fat diet induced NAFLD mouse model. In this work, we revealed the improvement of insulin resistance and lipid accumulation of four individual SGs from loquat leaves, including three known SGs (SG1, SG2, SG3) (*Ao et al., 2015*; *Chen et al., 2008*) and one new compound SG4. Furthermore, possible cellular mechanism of SGs on the amelioration of lipid-induced IR was elucidated by exploring downstream signal pathways involved in AMPK-mediated pathway.

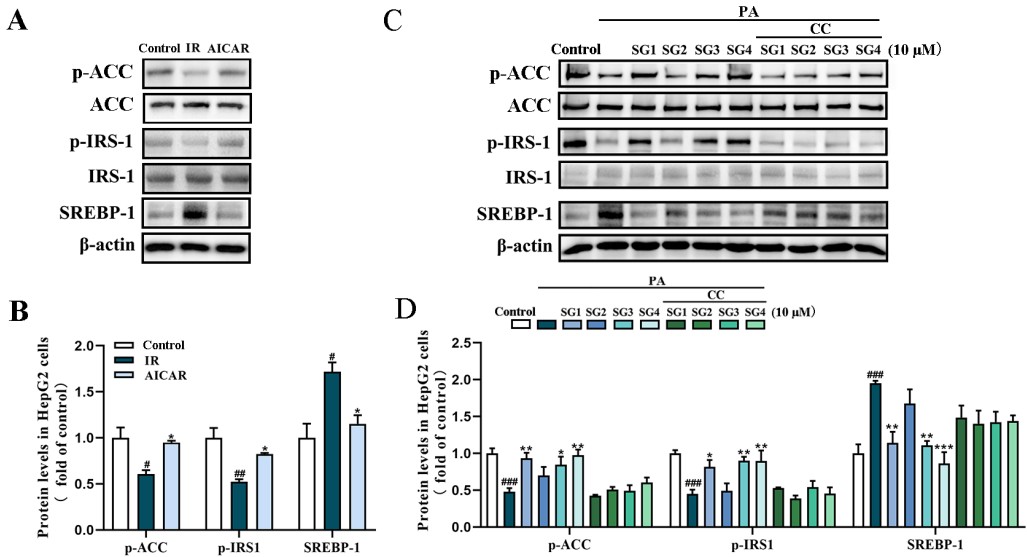

**Figure 5** **Western blot analysis of protein expression levels of AMPK signaling pathway after AICAR and CC treatment.** (A) Expression strips of the proteins of p-IRS1, IRS1, SREBP-1 and $\beta$-actin; (B) phosphorylation level of ACC normalized by ACC, phosphorylation level of IRS-1 normalized by IRS-1 and SREBP-1 protein level normalized by $\beta$-actin ($n = 3$) in HepG2 cells after AICAR treatment; (C) Expression strips of the proteins of p-ACC, ACC, p-IRS-1, IRS-1, SREBP-1 and $\beta$-actin; (D) phosphorylation level of ACC normalized by ACC, phosphorylation level of IRS-1 normalized by IRS-1 and SREBP-1 protein level normalized by $\beta$-actin ($n = 3$) in HepG2 cells after SGs and AICAR treatment. [###] $P < 0.001$ compared with the control group; [*] $P < 0.05$, [**] $P < 0.01$, [***] $P < 0.001$ compared with the IR group.

It is well known that IR is the main mechanism of T2DM pathogenesis. Liver is a critical metabolic organ in maintenance of glucose and lipid homeostasis (*Birkenfeld & Shulman, 2014*). It has been shown that excessive nutrients induce a greater increase in liver fat and insulin resistance and long-term treatment with PA can cause IR (*Malik et al., 2019*). PA-induced cells have been widely used as an IR model *in vitro*. In the present study, we used 0.25 mM PA to induce IR in HepG2 cells. The CCK-8 assay showed that 5 μM or 10 μM SGs were safe in IR HepG2 cells. Further, we found that SGs significantly enhanced glucose uptake in PA-induced IR HepG2 cells. On the other hand, SGs administration established a significant decrease of lipid deposition in PA-induced HepG2 cells, as well as down-regulated intracellular TC and TG levels at a low concentration.

AMPK, an energy sensor, is believed to systematically adjust the lipid metabolism balance, including the regulation of plasma glucose levels, fatty acid oxidation and glycogen metabolism. Several evidences indicated that activation of AMPK is an effective approach for improving insulin sensitivity by stimulating glucose uptake (*Fryer & Carling, 2005*). In the liver, AMPK activation promotes glucose uptake and suppresses lipid and cholesterol synthesis (*Yan et al., 2018*). Acetyl-CoA carboxylase (ACC) is a key substrate of AMPK (*Cesquini et al., 2008*). Once activated, AMPK modulates insulin signaling through the phosphorylation of key proteins that promotes insulin responsiveness, including IRS and Akt. The dysregulation of IRS is a common underlying mechanism in IR (*DeFronzo & Tripathy, 2009*; *Karlsson & Zierath, 2007*). Among isoforms of IRS, IRS-1 is more

closely related to glucose homeostasis. Two forms of phosphorylation of IRS-1, serine phosphorylation and tyrosine phosphorylation, coordinately regulate insulin signaling. It has been reported that the serine phosphorylation of IRS-1 protects IRS-1 from the tyrosine phosphorylation under insulin stimulation (*Gual, Le Marchand-Brustel & Tanti, 2005*). In addition, the serine phosphorylation of IRS-1 inhibits Akt activity under obese conditions and then impairs IRS-1/PI3K/Akt signal pathways (*Gao et al., 2002*). Hence, we investigated the effects of SGs on the expression of the target genes associated with insulin signal pathways. In this study, PA reduced the phosphorylation expression of AMPK, ACC, IRS-1 and Akt. In consist with the up-regulation of glucose uptake, SGs reversed the phosphorylation expression of AMPK, IRS-1 and Akt in PA-induced IR HepG2 cells. PA-induced decrease of ACC and IRS1 phosphorylation was reversed by AICAR. Importantly, SGs mediated up-regulative effect on ACC and IRS1 phosphorylation was partly eliminated by CC. These results demonstrated that SGs could ameliorated insulin resistance through AMPK/IRS-1/Akt pathway.

Sterol regulatory element binding proteins (SREBPs), are key lipogenic transcription factors regulating cellular lipid metabolism and modulated by glucose and insulin (*Ruiz et al., 2014*; *Zhu et al., 2019*). Among isoforms of SREBPs, SREBP-1 is essential to control expression of enzymes of carbohydrate and fatty acid metabolism (*Hua et al., 2016*). It was well-documented that SREBP-1 positively regulated the expression of genes encoding lipogenic enzymes including fatty acid synthase (FAS) (*Shimano, 2001*). As a cellular energy sensor, AMPK signaling is regarded as one of the most important regulators of lipid homeostasis (*Hardie, Ross & Hawley, 2012*). On the upstream of SREBP-1, AMPK could suppress SREBP-1 mediated lipogenesis (*Li et al., 2011*). As expected, in consist with the down-regulation of intracellular TC and TG levels and the alleviation of lipid deposition, SGs down-regulated SREBP-1 and FAS protein level in PA-induced IR HepG2 cells. In addition, AICAR reversed PA-induced SREBP-1 elevation, while CC partly abolished SGs mediated SREBP-1 reduction. These results demonstrated that SGs repressed the lipogenesis in IR HepG2 cells via the AMPK/SREBP-1/FAS pathway.

Noticeably, SG4 was the most prominent compound when comparing to other SGs, while SG2 had no significant effect on insulin resistance improvement. Compared with SG1, SG3 had a similar therapeutic effect. By comparison of structure–activity relationship, we found that SGs were composed of nerolidol as aglycone and oligosaccharide chain. It is suggesting that the oligosaccharide chain part influences the effect, and the difference among them occurs because of the saccharide category and the amount of saccharide. Compared with SG1, SG2 lacks one rhamnose on the junction with glucose C-6, while SG3 lacks the other rhamnose on the junction with rhamnose C-4 which is connected to glucose C-2. However, one rhamnose at the end of the oligosaccharide chain is replaced by arabinose in SG4. Based on the above analysis, we speculate that the number of saccharides in SGs unit has a significant effect on activities. Moreover, both C-6 of glucose and C-4 of rhamnose are potential active sites, which need to be further investigated.

## CONCLUSIONS

In summary, we stated a new sesquiterpene glycoside (SG4), namely as nerolidol-3-O-$\alpha$-L-arabinopyranosyl-(1 →4)-$\alpha$-L-rhamnopyranosyl-(1 →2)-[$\alpha$-L-rhamnopyranosyl-(1 →6)]-$\beta$-d-glucopyranoside, which was isolated in loquat leaf for the first time. Interestingly, our further research demonstrated that SG4, like other SGs from loquat leaf, could promote glucose uptake and ameliorate lipid accumulation in PA-induced insulin-resistant HepG2 cells. SGs may act as potential natural products regulating glucose and lipid metabolism in T2DM. The potential mechanism might be associated with the activation of the AMPK/IRS-1/Akt pathway and inhibition of the SREBP-1/FAS pathway.

### Funding

This work was supported by the National Natural Science Foundation of China (No.81773885; No.81703224; No.31770366; No.81973463) and the Jiangsu Scientific and Technological Innovations Platform (Jiangsu Provincial Service Center for Antidiabetic Drug Screening). The funders had no role in study design, data collection and analysis, decision to publish, or preparation of the manuscript.

### Grant Disclosures

The following grant information was disclosed by the authors:
National Natural Science Foundation of China: No.81773885, No.81703224, No.31770366, No.81973463.
Jiangsu Scientific and Technological Innovations Platform (Jiangsu Provincial Service Center for Antidiabetic Drug Screening).

### Competing Interests

The authors declare there are no competing interests.

### Author Contributions

- Jiawei Li conceived and designed the experiments, performed the experiments, analyzed the data, prepared figures and/or tables, authored or reviewed drafts of the paper, and approved the final draft.
- Xiaoqin Ding conceived and designed the experiments, prepared figures and/or tables, authored or reviewed drafts of the paper, and approved the final draft.
- Tunyu Jian conceived and designed the experiments, analyzed the data, prepared figures and/or tables, authored or reviewed drafts of the paper, and approved the final draft.
- Han Lü, Lei Zhao and Jing Li performed the experiments, prepared figures and/or tables, and approved the final draft.
- Yan Liu and Bingru Ren performed the experiments, authored or reviewed drafts of the paper, and approved the final draft.
- Jian Chen analyzed the data, authored or reviewed drafts of the paper, and approved the final draft.
## Data Availability

    The raw measurements are available in the Supplementary Files.

## Supplemental Information

Supplemental information for this article can be found online at http://dx.doi.org/10.7717/peerj.10413#supplemental-information.

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
