# Peer review of "Four sesquiterpene glycosides from loquat (Eriobotrya japonica) leaf ameliorates palmitic acid-induced insulin resistance and lipid accumulation in HepG2 Cells via AMPK signaling pathway"

_PeerJ, doi:10.7717/peerj.10413_

## Round 0.1 · original submission · Major Revisions

Three of the reviewers were very positive; each has a list of issue to address but mostly these represent relatively minor changes of English, or style. They should not present a problem. Reviewer-4 is less positive - please address points 2, 3 and 5 of reviewer-4 in the Discussion of your manuscript; they do not require extra work but should be highlighted so that the readership can make their own judgments. I shall deal with his/her other points below.

Hence I ask for the following which likely constitutes extra experimental work:

[1] You need to address point-1 of reviewer-4: How do we know insulin resistance has been achieved? It is not sufficient to replicate conditions from other studies - you need to show palmitate has induced insulin resistance in this set of experiments. This is essential. If you have used this approach in other studies in this cell line from YOUR lab, then the information in support of this must be clearly stated and signposted.

[2] An activator of AMPK should be tested in parallel in both the Palmitate/ethanol and control.

[3] Controls to dissect the effects of ethanol from palmitate are essential (reviewer-2). This is very important given the comments of the reviewer and the published confusion in the area.

[4] AMPK downstream substrates should be assayed to validate the conclusions.

I recognise this is extra work, but think the manuscript needs this for the conclusions to be robust. For this reason, I have noted this as 'Major revisions' but hope you can take this on-board and address these points.

Reviewer 1 ·

Basic reporting

Results are of interest to those working in this area of research. A clear report was presented.

Experimental design

Sound.

Validity of the findings

OK.

Additional comments

The reported influence of SG's on glucose uptake and lipid accumulation in insulin-resistant HepG2 cells is well-presented. As Eriobotrya japonica is often used in the traditional medicine of China, it is recommended to refer to a review on the use of it against diabetes [J. Ethnopharm., 92, 1-21 (2004)].
Some linguistic corrections need to be performed:
- line 81: delete 'natural'
- line 186: 'In the H1-...'
- line 187-188: 'all corresponded to ..'In addition, use superscripts for R)substituents in the formulae.

·

Basic reporting

This MS by Dr. Li and colleagues reports on the effects of sesquiterpene glycosides (SGs) on palmitic acid induced insulin resistance in HepG2 cells via AMPK signaling pathway. The experimental model involves treatment of HepG2 cells with palmitate + ethanol + BSA or with BSA alone. There does not seem to be challenge with insulin at any stage.
It has been reported previously that ethanol inhibits AMPK (You M, matsumoto M. et al Gastroenterology 2002; 127:1798-1808) but palmitate activates AMPK (Kawaguchi T. et al., JBC 2002, 277; 3829-3836). The authors therefore need to consider whether the decrease in AMPK by palmitate/ethanol (Fig. 4B) is due to the ethanol rather than the palmitate and whether the activation of AMPK by SG1-4 is independent of reversal of insulin resistance but is due to direct activation of AMPK. The model proposes that the activation of AMPK by SGs is upstream of IRS and Akt signalling. This can be tested using an allosteric AMPK activator such as A-769662. This would clarify whether the increase in glucose uptake (presumably glucose phosphorylation) is secondary to activation of AMPK or other mechanisms.


Basic Reporting:

Clear unambiguous except for the lack of clarity whether the model can be described as insulin resistant if: (i) insulin was not tested; (ii) whether the trigger was palmitate or palmiate plus ethanol, with the palmitate responsible for the TG and cholesterol storage and the ethanol for inhibition of AMPK as shown by Crabb DW et al 2004 Gastroenterology; 2008, Am J Physio).

Experimental design

Experimental design: If AMPK is considered to be upstream of IRS1, AKT (See Fig. 5) then an activator of AMPK should be tested in parallel in both the Palmitate/ethanol and control.

Authors should consider revising title to SGs ameliorate palmitate induced lipid accumulation and IRS signalling.

Fig. 3C and 3D are both labelled TC presumably, Fig-3D should be TG

Validity of the findings

The SGs appear to cause activation of AMPK. They have been tested in conjunction with palmitate and ethanol. The latter has been shown to inhibit AMPK. The SGs may be effective in conditions of lipid overload and attenuated AMPK. It would be of interest to test them with palmitate alone (without ethanol). A palmitate solution can be prepared without ethanol by warming 20mM sodium palmitate in water and then diluting it in cold BSA.

Additional comments

This MS by Dr. Li and colleagues reports on the effects of sesquiterpene glycosides (SGs) on palmitic acid induced insulin resistance in HepG2 cells via AMPK signaling pathway. The experimental model involves treatment of HepG2 cells with palmitate + ethanol + BSA or with BSA alone. It demonstrates that SGs activate AMPK and IRS>AKt signalling. Whether phosphorylation of IRS1 is downstream of activation of AMPK could be tested with an AMPK activator.

It has been reported previously that ethanol inhibits AMPK (You M, matsumoto M. et al Gastroenterology 2002; 127:1798-1808) whereas palmitate activates AMPK (Kawaguchi T. et al., JBC 2002, 277; 3829-3836). The authors therefore need to consider whether the decrease in AMPK by palmitate/ethanol (Fig. 4B) is due to the ethanol rather than the palmitate and whether the activation of AMPK by SG1-4 is independent of reversal of insulin resistance. The model proposes that the activation of AMPK by SGs is upstream of IRS and Akt signalling. This is an assumption but can be tested using an allosteric AMPK activator such as A-769662. This would clarify whether the increase in glucose uptake (presumably glucose phosphorylation) is secondary to activation of AMPK or other mechanisms.

The SGs may be effective in conditions of lipid overload and attenuated AMPK. It would be of interest to test them with palmitate alone (without ethanol). A palmitate solution can be prepared without ethanol by warming 20mM sodium palmitate in water and then diluting it in cold BSA.

Fig. 3C and 3D are both labelled TC presumably, Fig-3D should be TG

Reviewer 3 ·

Basic reporting

No comment

Experimental design

No comment

Validity of the findings

No comment

Additional comments

Their Editor,
The article concerns the isolation and structural elucidation of four sesquiterpene glycosides from loquat (Eriobotryajaponica) leaf. It evaluates the beneficial effects of these four single SGs isolated from loquat leaf, including SG1, SG2, SG3 and one novel compound SG4 against palmitic acid-induced insulin resistance in HepG2 cells. SG1, SG3 and SG4 could significantly enhance glucose uptake of insulin-resistant HepG2 cells at non-cytotoxic concentration.
The article is interesting and very well written but several imperfections should be corrected.

1) “D” and “L” used in IUPAC name should be lower size
2) δH and δC should be used instead of δ and should be italicized.
3) The carbon should be rewritting using one decimal instead of two.
4) Described acid hydrolysis and sugar analysis.
5) Why was the extraction and isolation done for two months.
6) In table one, number shoulb be put under the abreviations of sugars.
7) For abreviation of glucose, Glc should be use instead of Glu.
8) Rha I and Rha 2 should be used as abreviation for Rhamnose in bot the table and the text.
9) “ . ” should be used instead of “ , ” In table one in the Coupling constant of carbon two .
10) The coupling constant should be rewritting in the one decimal place.
11) In line 200-201, the correlation is in the HMQC spectrum.

Yours Sincerely

Reviewer 4 ·

Basic reporting

Figure 3 C D has some typos. I supposed that Figure 3C should be "TG" content and Figure 3D should be "TC" content. However, two figures had the same labels.

Experimental design

In general, the experimental design could be improved in following aspect.
1. When insulin resistance is claimed, there should be evidence showing that insulin signaling is impaired by palmitate treatment. Therefore, in addition to basal condition, insulin stimulated condition should be also considered. This part of the data was not shown in this manuscript.
2. Basal glucose uptake was reduced by PA treatment. Is this due to "insulin resistance" in the absence of insulin stimulation? Besides, the rationale to put glucose uptake data with TG and TC content is not clear. Glucose uptake should be related to glycogen deposition and separated from PA-induced lipid accumulation issue.
3. How do SGs sensitize PA induced insulin resistance via AMPK activation remained to be clarified. Inhibitor study should be employed.
4. Due to the observation that SGs compounds can activate AMPK and reduce TG and TC content in PA treated cells, AMPK down stream substrate should be also analyzed. Acetyl-CoA carboxylase (ACC) is one of the key substrate for AMPK to regulate beta-oxidation. Therefore, effects of SGs on ACC phosphorylation should be also measured. Such effect should be blocked by AMPK inhibitors to demonstrate that SGs action is really mediated by AMPK activation.
5. SREBP1 downstream genes related to lipid metabolism should be also measured in the current study to claim that SGs inhibits lipid deposition via SREBP1 .

Validity of the findings

no comments

Additional comments

In general, findings are too preliminary and experiments need to be expended. Insulin stimulation condition should be included in "Insulin resistance" study. Inhibitor for AMPK should be used to strengthen the validity of finding in the investigation.

---

## Round 0.2 · accepted · Accept

Thank you for addressing all of the points raised by the first round of review.

Reviewer 1 ·

Basic reporting

The manuscript was sufficiently corrected and adapted.

Experimental design

-

Validity of the findings

-

Additional comments

The manuscript was sufficiently corrected and adapted.